# A Qualitative Study of the Benefits and Utility of Brief Motivational Interviewing to Reduce Sexually Transmitted Infections among Men Who Have Sex with Men

**DOI:** 10.3390/bs13080654

**Published:** 2023-08-04

**Authors:** Matshidiso A. Malefo, Olalekan A. Ayo-Yusuf, Mathildah Mpata Mokgatle

**Affiliations:** 1School of Public Health, Sefako Makgatho Health Sciences University, Pretoria 0204, South Africa; mathildah.mokgatle@smu.ac.za; 2School of Health Systems and Public Health, University of Pretoria, Pretoria 0028, South Africa; lekan.ayo-yusuf@up.ac.za

**Keywords:** men who have sex with men, sexually transmitted infections, brief motivational interview, benefits, utility, qualitative study

## Abstract

Several studies have demonstrated the effectiveness of motivational interviewing (MI) in reducing sexual risk behaviors. However, limited information is available on the acceptability of brief MI among men who have sex with men (MSM) in poor resource settings like sub-Saharan Africa. The objective of this study was to assess the views of MSM about the benefits and utility of brief MI (bMI) in changing their risky behavior. A qualitative study among men who have sex with men (MSM) who were enrolled in a longitudinal observational study between December 2021 and May 2023. The setting was in Tshwane North and participants were scheduled for baseline, 6-month, and 12-month visits. All participants received 20 min one-on-one face-to-face brief motivational interview (bMI) sessions during their follow-up visits. At month 12, an exit interview was conducted with consenting conveniently sampled participants (n = 23) who had completed all scheduled visits and received three bMI sessions. The findings indicated that the most recalled conversation was related to multiple sexual partners, having sex under the influence of alcohol, and MSM learned more about sexually transmitted diseases. Many expressed being comfortable with the sessions because the counselor was respectful and non-judgmental. Most found the bMI sessions to have a positive impact on changing and reducing risky sexual behaviors, particularly it reportedly increased their use of condoms and reduced the number of multiple partners. MSM found the bMI to be useful and acceptable in reducing sexual risk behaviors among MSM.

## 1. Introduction

Men who have sex with men (MSM) constitute a risk group for STIs and are at greater risk of poor mental health and substance use than the general population [1,2]. Engagement in sexual risk behaviors such as having multiple partnerships, sexual concurrency, sex under the influence of drugs or alcohol, and interchange sex is common among young people and MSM, which can contribute to high rates of HIV and sexually transmitted infections (STIs) [3,4,5]. As part of a wide-ranging strategy for HIV prevention and care, behavioral interventions are an important tool in the global fight against HIV/STIs [6]. Increased HIV pre-exposure prophylaxis (PrEP) initiation, and adherence can help end the epidemic because it significantly reduces acquisition risk among vulnerable individuals [7,8,9]. However, PrEP use remains minimal among MSM due to different factors such as medication stigma, discomfort discussing sexual risk behaviors with clinicians, and perceived low HIV risk [10,11,12]. Cost-effective interventions such as motivational interviewing (MI) [13,14] could prevent barriers, reduce inequalities, and improve PrEP initiation and adherence.

One-to-one behavioral interventions such as motivational interviewing (MI) have been suggested to reduce HIV in high-risk groups [15,16].

MI is a brief intervention that is used to address a variety of health behavior issues [17]. It is a client-centered counseling technique that has been successfully utilized to promote behavior change by eliciting client motivations for change [18]. MI is a counseling technique that helps patients to explore and resolve their ambivalence toward changing their behavior [19]. According to Naar-king and colleagues [20], MI interventions are effective in changing behavior because they create a clear context for improving client communication and promoting client-centered behavior change with the aim of increasing motivation and self-efficacy. Several studies, particularly in the United States of America, indicate that MI intervention can considerably reduce risky sexual behavior and boost condom use [21,22]. Some behavioral HIV prevention interventions utilize MI because of its benefits [23,24,25]. MI techniques are often used in health promotion because of their cost-effectiveness, adaptability, and use of client interests for behavior change. MI started with addiction counseling and has been utilized as an intervention approach globally to support effective behavior change for many health behaviors [26], including dieting [27], exercise [28], violence prevention [29], and medication adherence for chronic conditions [30].

A study in Thailand showed that motivational interviewing could be used to improve adherence to alcohol usage and antiviral medication [31]. Interventions based on the principles of motivational interviewing have been shown in the United States to significantly decrease high-risk sexual behavior and increase condom use [21]. Another study highlighted that combined motivational interviewing and cognitive behavioral approaches decreased alcohol use [32]. MI has been demonstrated to be effective in reducing drug use and improving sexual health in adolescents—predominantly in sexual minority adolescents [33]. Evidence to date shows using MI is more effective in changing patients’ behaviors and improving their oral health compared to using traditional health education approaches that focus on disseminating information and giving normative advice [34,35]. A systematic review by Kay et al. [36] found MI to potentially help patients to improve their oral health by increasing patients’ oral health knowledge and subsequently their oral health behaviors. Another systematic review by Carra et al. [34] demonstrated that MI promoted positive behavior change and improved the oral health of patients with periodontal diseases. Examples of those improvements include decreased plaque levels, decreased bleeding upon probing, increased toothbrushing, and increased interdental cleaning.

Studies examining the benefits and effectiveness of MI on STI and HIV prevention behaviors among MSM in Gauten region, Tshwane North were limited. There is a need to group scientific research with a high level of the effectiveness of interventions using MI to reduce risk behaviors among high-risk populations. Furthermore, knowledge about motivational interviewing can help nurses and other health professionals in assisting MSM in reducing their risky behavior practices, PreP use, and adherence to ART medications. This study therefore sought to assess the views of MSM about the benefit and usefulness of the brief MI (bMI) they had received in changing their risky behavior.

## 2. Methodology

### 2.1. Study Design

This was a qualitative descriptive study as it intended to explore the benefits and utility of bMI among the MSM who received this intervention as part of a clinical trial.

### 2.2. Study Population and Setting

The population of the study consisted of MSM who reside in the area of Tshwane North District and received bMI as part of a clinical trial. This study enrolled 200 MSM and participants were followed up at 6- and 12-month intervals. During baseline and scheduled follow-up visits, all MSM received 20-min bMI sessions. This study was conducted at the Medunsa Clinical Research Unit (MeCRU) in Tshwane North District. MeCRU is an established research unit appropriate to assist with recruiting MSM from the communities and testing them. The site was also established for research that involves tests for HIV and STIs (rectal, urethral, and oral). This study consisted of MSM who had completed all three bMI sessions and their twelve-month visit.

### 2.3. Sampling Technique and Size

The purposive sampling of MSM who had gone through three sessions of bMI and completed their twelve-month visit in a clinical trial was performed. The sample size was determined (n = 23), following a previously suggested guideline [37] and informed by the principle of data saturation, where no new ideas are emerging during the performance of the research.

### 2.4. Recruitment, Data Collection Methods, and Tools

Participants were recruited from the main study MSM sample that took up bMI sessions. Upon recruitment, the informed consent process included introduction of the researcher to the participants, goals and assumptions of the researcher, description of the study aims, objectives, the intended use of study results. The researcher used an interview guide containing open-ended, semi-structured questions to garner a deeper insight into the thoughts and views of the participant about the acceptability and usefulness of MIs. Face-to face interview was conducted and the duration of the interviews ranged from 15–30 min. All semi-structured interviews were undertaken by researcher MAM. She was a female, aged 46 at the time of the interviews. She had more than 14 years of social behavioral experience with qualitative research. No relationship with MAM was gained prior to the interview. A tape recorder was used to record all interviews. The interviews were transcribed verbatim by the researcher so that the transcripts would be as accurate as possible and understandable to the reader [38]. Informed consent forms were signed at the beginning of the study before any procedures were undertaken with the participants. The MI exit interview questionnaire guide covered the following: How did you feel about the motivational interviews? What topic or priority risky behavior did you speak about? What made you choose that specific topic? How helpful or useful was the MI? What did you like about the MI? What is it that you did not like about MI sessions? Do you think you or other MSM will benefit from MI session? Why? What changes occurred in your life after receiving MI sessions? What can be undertaken to improve MI sessions?

### 2.5. Motivation Interview

The MI sessions were based on the top ten priority risky sexual behaviors (both receptive and insertive anal sex, having sex with multiple partners, having sex with an HIV-positive partner without a condom, having sex with a partner of unknown HIV status, group sex, both receiving and performing oral sex, having sex without a condom, having sex under the influence of alcohol, both performing and receiving rimming, sharing sex toys) previously identified by the MSM experts [39]. MI is an approach in which a counselor helps a client to explore the client’s motivation and confidence to change his/her risky behaviors and to identify suitable ways to deal with obstacles to such behavioral change [19]. The five guiding principles of MI are (a) expressing empathy, (b) developing discrepancy, (c) avoiding argumentation, (d) rolling with resistance, and (e) supporting self-efficacy [40]. We opted for a brief MI approach [41] as the sessions had to be feasible in dealing with MSM’s risky sexual behaviors. The suggested duration of an MI session was 20 min.

In the first MI session, the counselor established rapport and determined the participant’s risky behaviors. With the support of the counselor, the participant then identified his priority problem for change, thus setting the agenda for change. This agenda for change might include a plan to reduce or stop practicing risky behavior. The second MI session built on the previous session, dealt with the challenges relating to the previously set agenda, and then proceeded to the next behavioral problem. The third session either dealt with the final problem identified or reinforced the previous message.

### 2.6. Data Analysis and Coding

Recorded exit interviews of a sample of n = 23 were purposefully selected for data coding and analysis. The first step was transcribing the audio recording verbatim and translating the Setswana audio into English. Two authors are Setswana-speaking, and the co-author is English-speaking. Data in the form of an audio recording were loaded into Microsoft Office 365 to translate them into English to be able to listen to the audio often without losing its meaning. The researchers analyzed the data using thematic content analysis. Data were read repeatedly and thoroughly to develop initial common codes. The researchers used the initial codes, categorized them into groups, and assigned labels in the form of themes. Further relations and sub-groups emerged from the data and were placed under the umbrella themes to form sub-themes. The organized data resulted in a codebook, which together with the 23 transcripts, were uploaded into QSR NVivo version 12 software for analysis. Trustworthiness was considered by the use of reflexivity by the researcher by adhering to the COREQ guidelines [42] and by documenting the follow strategies to enhance rigor [43]: (a) an audit trail, (b) attention to negative cases (searching for instances when participants directly contrasted or contradicted the themes and content identified) within the analysis, (c) peer debriefing (MAM presented a defendable case to other authors OAA, MMM) for critical comment after 17 interviews. Further to this, MAM was required to provide a defendable case to the research group about the findings. The presentation of the findings to the group was made after interview 23. Following this, interviews were focused on exploring the themes and saturating content identified. Quotes are presented verbatim and identified by a participant number, e.g., PPT 01.

## 3. Results

### 3.1. Demographic Characteristics of Participants

Of the study participants, the majority were within the age range 20–30 years old (n = 15), had completed high school (n = 15), were in a casual relationship (n = 15), and all participants reported that they were not married at the time of the data collection. Of the participants, only four were employed and reported practicing anal sex (n = 10). Other characteristics of the participants are displayed in Table 1.

Thematic content analysis was used, and the following themes and sub-themes emerged during the data analysis (see Table 2).

### 3.2. Theme 1: Motivation for MSM to Join the Study

The participants reported that they decided to participate in the study because they wanted to know about their health status. They do not regularly visit the clinics just to test for STIs unless they are sick. They indicated that it was beneficial for them to be tested for asymptomatic STIs and to be treated immediately. This is explained in the quotations below:


*“What motivated me to participate in this research was that sometimes you have to know your health status you know I am gay and we have a lot of sex and I don’t use condom so you can have diseases that I am not aware of so it is best to join research so that you will know your health status.”*
(PPT 02)


*“What encouraged me is that this research can discover things that I am not aware of like those STIs and my health status. Even when you visit the clinic especially us gays, they will not test those STIs that your research tested. They only focus on the ordinary STIs. They don’t go deeper like you.”*
(PPT 05)


*“I wanted to know if I have STIs or not, and I also wanted to know my health status.”*
(PPT 17)


*“Mmmm… I wanted to learn more about the study and if it wasn’t for this study by now I will still be having the STIs.”*
(PPT 03)

Only one participant indicated that the other purpose for him joining the study was also re-imbursement with money.


*“Mmmm… The reason I joined the study was I wanted to know if I have STIs or not and also money re-imbursement.”*
(PPT 08)

### 3.3. Theme 2: Usefulness of bMI Sessions for Behavior Change among the MSM

After the third MI session, which was 12 months back, we were able to gauge the participants’ perceptions and views about the usefulness of the sessions. The participants claimed that the sessions allowed them to understand the current state of their risky behavior. Phrases such as “I see, tell me more,” helped the participants to reflect deeply about their response. Some participants indicated that they had managed to reduce some of the risky behaviors that they had been practicing. Most participants benefited from receiving the MI sessions.


*“It was helpful eish… I reduced sleeping around with different men.”*
(PPT 04)


*“It was helpful because now I can be able to negotiate condom use with my partner. I am no longer having group sex. I can even encourage my partner to go and test for HIV.”*
(PPT 05)


*“It was important because I decided to break up with my partner. I wanted to focus on myself and my wellbeing. The counselor was also open, and you can trust her because of confidentiality and she treat us with respect.”*
(PPT 08)


*“The session was helpful but not that much. I have reduced having sex under the influence of alcohol without using any protection. At least now I am trying to use protection than before.”*
(PPT 20)


*“I benefited because I have reduced number of sex partners from 5 to 2. I can take control of my sex life and after that now I feel good about myself.”*
(PPT 21)

Participants were asked what their views and perceptions were about recommending MI sessions to other MSM who had not participated in the study. Most participants indicated that they thought MI sessions would be beneficial to other MSM.


*“Yes they will benefit because they can maybe start using condoms, not to do group sex especially when they are drunk.”*
(PPT 06)


*“Yes they will benefit like by being told about those STIs I didn’t know about them so by knowing will help them to take care of their health and behavior and they can start using protection. Once you saw those STIs you will definitely change your behavior.”*
(PPT 12)


*“Yes, a lot of they will benefit knowledge about the risky life that they are doing. They can start by reducing number of sex partners, try to use condoms and reducing having sex under the influence of alcohol.”*
(PPT 22)

Most participants mentioned that after receiving MI sessions, they had managed to change their risky behavior.


*“Sometimes I was not using condoms but now since participating here I am using condoms and I am no longer having sex with anyone in the taverns—you know, the one night stands.”*
(PPT 01)


*“Before I was just living. I didn’t care sleeping with different men, changing them, maybe 4–5 different men a week, but after receiving counseling I have changed my behavior. I reduced the number of partners and I start using condoms and I am focused now.”*
(PPT 03)


*“I have changed. I am no longer behaving like I used to do before. Tjhoo… I was reckless, sleeping around, so after counseling sessions I have changed.”*
(PPT 04)


*“It has changed because now I have one partner and I am reduced this thing of sleeping around under the influence of alcohol and I use protection.”*
(PPT 09)

### 3.4. Theme 3: Reasons for Choosing Topics for Discussion by MSM

During the MI sessions, the participants had to set the agenda for the risky behavior topics they would prefer to talk about. Most of the participants spoke about the risk of having multiple partners, the risk of not using condoms, having sex under the influence of alcohol, having sex in exchange for money, and having group sex.


*“We spoke about multiple partners, having unprotected sex with unknown HIV-positive person, group sex. We also spoke about having sex under the influence of alcohol.”*
(PPT 11)


*“We spoke about how to prevent STIs, multiple partners, having sex under the influence of alcohol. We spoke about a lot.”*
(PPT 12)


*“We spoke about a lot of things, I can’t remember, but I know that we spoke about the risk of having multiple partners and not using condoms, having sex in exchange for money and under the influence of alcohol.”*
(PPT 14)


*“We spoke about a lot of things like the risk of having multiple partners, group sex and having sex under the influence of alcohol.”*
(PPT 17)

Some participants also gave reasons why it was important for them to talk about those topics during the MI sessions.


*“Because those are everyday topic things that we practice every day, so you need to be comfortable and honest. When you are not honest you are not going to get help.”*
(PPT 02)


*“It was important because if you went through that experience of maybe having group sex under the influence of alcohol and you didn’t use any protection it is a mistake that you can talk about, so you don’t repeat that mistake again. Sometimes also a pressure from your partner, both of you were using condoms, then things start to change where he doesn’t want to use condoms anymore during sex. Talking about them makes you feel better. It is like you off-load.”*
(PPT 05)


*“Because I wanted to learn more. You know, you cannot ask or talk to someone you are not comfortable with. Some people are judgmental. So, I was free to talk about such because she was not going to tell anyone about what we have discussed.”*
(PPT 07)

One participant mentioned that during the MI session, they also spoke about challenges that gay populations are facing daily in their communities.


*“Eish… We spoke about the risk of having multiple partners, gay sex life and practices, challenges that gay people are facing in the community. It is not everyone who will accept us as gays. Some people discriminate and judge us. Some is due to lack of knowledge.”*
(PPT 05)

### 3.5. Theme 4: Acceptability of bMI Sessions among the MSM

Most participants said that they liked the MI sessions. They had been open and had felt respected. It had been easy for them to be free and to talk openly about their sex lives and their risky behaviors.


*“What I liked more with counseling, a lot of things, things that I didn’t know them and not being judged or stigmatized. So that is why I was open to talk about anything.”*
(PPT 01)


*“It made me to notice my mistakes and behavior and now I have changed my wrong behavior. I am trying to be a better person.”*
(PPT 05)


*“It made me to be more open about my sexuality and personal things that I was going through.”*
(PPT 10)


*”I liked everything. The counselor was open and friendly and made me talk more about my sex life and things that are happening around me. We know that we are living a risky life but having someone showing you that this lifestyle can be changed, it is a good thing.”*
(PPT 15)


*“I liked the sessions because now I can talk to my friends about risk behaviors.”*
(PPT 19)


*Participants indicated that their overall experience of MI sessions was informative.*



*“Everything was fine. I have learned a lot of things like those STIs, the risky factors of having sex in exchange of money, and having sex under the influence of alcohol.”*
(PPT 09)


*“Everything was fine, because I discovered new information about STIs that I didn’t know that they exist.”*
(PPT 11)


*“What I liked more with that counseling is that talking to a neutral person, listening to you and trying to assist you to make better choices about your life it is good.”*
(PPT 21)


*“It was OK, informative and she was friendly so it was easy for me to talk to her.”*
(PPT 23)

One participant mentioned that he was uncomfortable talking about having sex in exchange for money, because it is a behavior that he engages in, especially when he is in need of money.

The participants were asked if there was anything that they did not like about the MI sessions, and most of them indicated that there was nothing that they did not like.


*“I was uncomfortable when she was asking me personal questions like having sex in exchange for money. I mean I don’t have a choice, especially when you don’t have money. You will do anything just to get money, even if you don’t like what you are doing.”*
(PPT 11)


*“Tjhoo… Nothing! Everything was OK! I was able to ask questions and together we were in the same page.”*
(PPT 07)

## 4. Discussion

This study aimed to assess the views of MSM about the usefulness and acceptability of MIs in changing risky behavior. This study found that MI counseling can provide solid examples of risk behavior reduction strategies and that the MSM receiving MI-based interventions demonstrated increased risk behavior reduction. This study showed that a brief three-session MI was useful, beneficial, and acceptable in reducing risky sexual behaviors among MSM over time in a Tshwane North context.

The fact that the study achieved the recruitment of the MSM through peer referrals and the findings revealed that there was a sense of motivation to join the study show the potential for improving health-care-seeking behaviors in key populations through the provision of dedicated services. According to reports by the MSM in this study, the attitude of the health-care service provider and being non-judgmental make the services acceptable and useful. Consistent with previous studies conducted among MSM, avoiding acts that disadvantage or discourage access to health care such as stigma, prejudice, discrimination, and other attitudes that display inequality toward MSM can improve access to health care for the key population [44,45]. Studies conducted among health-care service providers further conform that positive attitudes and interactions with MSM and other key populations improve access and use of services [46,47,48].

MI is a type of counseling that uses client-centered techniques to motivate an essential effort to change and to resolve uncertainty between present behavior and preferred outcomes [49]. The results of this study demonstrated that MSM benefited from receiving MI counseling and that MI was useful in changing their risky sexual behaviors. As found in the literature review conducted by Wulandari et al. [33] of studies regarding the effect of MI on adolescent risk behaviors, this study found that MI is beneficial in reducing risky sexual behavior in adults, which can potentially prevent HIV transmission in this high-risk group. The results are also similar to the (cis)male couples randomized control trial conducted by Stark et al. [50] in New York which demonstrated initial evidence of the potential for MI to address drug use and reduce sexual risk-taking behaviors among minority (cis)male couples at highest risk and the potential to produce benefits from the treatment options.

The participants in this study claimed that they had changed their sexual behavior after participating in MIs by starting to use condoms, and that they had reduced the frequency with which they had sex with multiple partners. This study’s findings support those of an earlier study by Herdiman et al. [51] which found that counseling grounded in motivational interviewing can illustrate effective strategies for lowering risky behaviors. Unprotected anal sex is a primary cause of HIV transmission among MSM. Our study found that MI can significantly improve condom use during anal sex. This is consistent with the behavioral prevention intervention studies conducted in the USA on MI among young MSM living with or without HIV [52,53].

Using motivational interviewing seems to have effectively increased condom use during anal sex, as with previous studies [52]. Our study findings are also consistent with research conducted in the United States of America which demonstrates that MI interventions can considerably reduce risky sexual behaviors and boost condom use [21]. Similarly, a randomized control trial conducted by Gold et al. [54] indicated that although MI is considered easier to utilize than informative educational counseling, the intervention has a significant effect, and can help reduce the incidence of unprotected sex, drug use, and HIV among research participants [55]. However, there was no consistency of condom use over time. As in a study conducted in China among MSM, there was no significant, consistent change in condom use for oral sex or in the number of sexual partners over time [52].

Our study found that MI sessions were acceptable to the MSM population because they reported that MI was informative and enabled them to discuss sensitive risky sexual behavior topics to reduce their risky behaviors. This is similar to a systematic review study of six randomized controlled trials targeting high-risk sexual behaviors among HIV-positive adults which showed that MI had the potential to reduce the incidence of high-risk sexual behaviors [20]. The results reported by Rongkavilit et al. [56] also found that MI could significantly reduce UAI among HIV-positive youth. The effectiveness of MI has been reported among different populations, in different settings, and for different problems [57,58,59]. This study confirms previous findings that HIV/STI prevention education using motivational interviewing can decrease sexual activity among MSM. Furthermore, this study shows that motivational interviewing counseling affects the increasing awareness of STI transmission prevention behavior in MSM. This study is in line with Ekasari et al. [60], who proved that structured awareness with counseling helped in developing positive changes in adolescents.

Effectively reducing high-risk sexual behaviors among MSM remains an important strategy for controlling the HIV/AIDS epidemic and lowering the incidence of HIV and sexually transmitted diseases.

## 5. Conclusions

BMI is beneficial, acceptable, and useful for reducing risky sexual behaviors in the MSM population and in preventing HIV/STI transmission. This study demonstrates that MI has an effect in moderating sexual behaviors in MSM, which is important in the prevention of HIV/STI transmission. The long-term effectiveness of MI conducted by trained health-care providers or counselors in improving consistent condom use during anal or oral sex among MSM needs to be evaluated in future studies.

Stigma and judgment by health workers toward people living with HIV (PLHIV) and key populations can undermine the uptake of HIV services. Planning and implementing targeted sensitization and stigma reduction interventions within health settings are critical to meet the needs of vulnerable populations that face more stigmatizing attitudes from health workers. Instead of employing a one-size-fits-all intervention for risky sexual behaviors, the findings of this study highlight the benefits of using MSM-led sexual education topics to enable intervening on patient-specific and relevant behaviors.

### Limitation

Participants in this study were chosen using purposeful sampling and may not be representative of all MSM in South Africa. Therefore, the findings of this study cannot be generalized to all MSM. This study employed a qualitative approach and, hence, was limited in measuring how the themes about the utility of MI improve various aspects of counseling MSM regarding sexual behavior, HIV, and STIs. The researchers recommend that future quantitative research using an analytical approach should be conducted. This study should not be considered as an exact documentation of behavior change. The results must be viewed as common realities experienced by participants. Specific implications were identified: (a) bMI could present a good supplementary intervention in healthcare facilities to support, promote motivation, personal accountability, and to develop goals for reducing sexual risk behavior in order to reduce the risk of STI transmission. By using motivational interviewing, a counselor may address each client’s unique barriers to change. (b) It is evident that bMI appeared to aid changes in perceptions of behavior and risky sexual activity. It was also useful and beneficial in reducing risky behaviors. This finding should be considered within health-care facilities as client-centered interventions play a role in allowing perceived changes and positive behavior to occur.

## Figures and Tables

**Table 1 behavsci-13-00654-t001:** Profile of study participants.

Demographic Information of the Twenty-Three (N = 23) Participants Who Were Interviewed	
**Age**	n (%)
20–30	15 (65.2)
31–40	7 (30.4)
**Highest level of schooling**	
Tertiary	8 (34.7)
High school	15 (65.2)
**Employment status**	
Employed	4 (17.3)
Unemployed	19 (82.6)
**Marital status**	
Single	23 (100)
**Type of relationship**	
Casual	15 (65.2)
Stable	8 (34.7)
**Sexual orientation**	
Gay	21 (91.3)
Bisexual	2 (8.6)
**Number of sexual partners in the last month**	
One	6 (26.0)
Two	5 (21.7)
Three or more	12 (52.1)
**Prevention method during sex**	
Condoms	12 (52.1)
Both condoms and lubricant	8 (34.7)
None	3 (13.0)
**Sexual practices**	
Anal sex	10 (43.4)
Anal sex and oral sex	8 (34.7)
Anal sex and rimming	3 (13.0)
Anal and vaginal sex	2 (8.6)

**Table 2 behavsci-13-00654-t002:** Themes and sub-themes.

**Motivation for MSM to Join the Study**
Sub-themes: **Reasons for joining the study.** **Encouragement for joining the study.** **Motivation for joining the study.** **Purposes of joining the stud.**
**Usefulness of bMI sessions for behavior change among the MSM**
Sub-themes: **Importance of MI sessions.** **Benefit of MI sessions.** **Helpfulness of MI sessions.** **Managed to change risky behavior.**
** Reasons of choosing topics for discussion by MSM **
Sub-themes: **Reasons for choosing those MI topics.** **Importance of talking about the chosen topics.**
**Acceptability of bMI sessions among the MSM**
Sub-themes: **Feeling of being respected and openness during MI session.** **MI sessions were informative.** **MI sessions enabled discussing sensitive risky sexual behavior topics.**

## Data Availability

Data will be made available upon request.

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
