# Peer review of "A Qualitative Study of the Benefits and Utility of Brief Motivational Interviewing to Reduce Sexually Transmitted Infections among Men Who Have Sex with Men"

_behavsci, 2023, doi:10.3390/bs13080654_

Round 1
Reviewer 1 Report
· Remove “penile” from STI tests on line 66.
· Table: please indicate period of time for number of sexual partners.
· For prevention method, was PrEP use assessed? Seropositioning?
· For sexual practices, were sexual positioning practices assessed (i.e., receptive vs insertive anal sex)?
· Please clarify your thematic analysis process. How were themes identified and what do you mean by “subthemes?” I think the themes should be reframed to illuminate the true meaning of the theme for example “Participating in an MI based study to learn more about their health” could be your first theme but the language use of “subthemes” seems inappropriate.
· Consider recent literature published in Prevention Science that MI-based interventions should be reframed given their persuasive nature in health promotion. Please consider incorporating this literature and perspectives in your discussion section:
Dangerfield II, D. T., Davis, G., Pandian, V., & Anderson, J. N. (2023). Using Motivational Interviewing to Increase HIV PrEP Initiation and Adherence: A Scoping Review. Prevention Science. https://doi.org/10.1007/s11121-023-01554-w
Author Response
RESPONSE TO REVIEWER 1 COMMENTS
Point 1: Remove “penile” from STI tests on line 66.
Response for point 1: Penile removed.
Point 2: Table: please indicate period of time for number of sexual partners.
Response for point 2: Corrected. It was number of sex partners in the last month.
Point 3: For prevention method, was PrEP use assessed? Seropositioning?
Response for point 3: No, options that were used for prevention methods were those indicated in the table but there was an option of other and nothing was indicated.
Point 4: For sexual practices, were sexual positioning practices assessed (i.e., receptive vs insertive anal sex)?
Response for point 4: No, that was not assessed. We were only asking about their sexual practices.
Point 5: Please clarify your thematic analysis process. How were themes identified and what do you mean by “subthemes?” I think the themes should be reframed to illuminate the true meaning of the theme for example “Participating in an MI based study to learn more about their health” could be your first theme but the language use of “subthemes” seems inappropriate.
Response for point 5: Development of themes is explained in line 159-167.
Point 6: Consider recent literature published in Prevention Science that MI-based interventions should be reframed given their persuasive nature in health promotion. Please consider incorporating this literature and perspectives in your discussion section:
Response for point 6: Literature incorporated as advised.
Reviewer 2 Report
Peer Review
Overall, the study was well-written, and I appreciated the authors’ subject matter. However, the introduction/background, rationale for study, contribution to science, and discussion are very underdeveloped and should be addressed before publication.
Abstract:
Says there is limited research on MSM and motivational interviewing. However, there is a lot of previous research on MSM and motivational interviewing. Please eliminate this statement and instead tell us how this study is different (-maybe with respect to this particular population or some other characteristic or outcome; maybe it’s the fact that this study specifically asked participants about the MI process, as opposed to delineating actual behavioral outcomes -I don’t know, but it needs to be clarified). This issue is further discussed in the Introduction feedback below as well.
Introduction:
Mentions that several studies have shown positive outcomes for the utility of MI in addressing risky sexual behavior, but only two studies are identified in the citations. Please add more studies to this citation.
Also, it would be beneficial to outline what some of the positive outcomes are that previous studies have found with motivational interviewing by reviewing two or three previous studies. What outcomes were the previous research authors looking to impact; describe the population they studied; what did they do; and what did they find regarding motivational interviewing and its impact on risky sexual behavior?
Also identify how your study would be different than previous research, as there is already much previous research on MSM and motivational interviewing. There is a 2011 study (Berg, Ross & Tikkanen, 2011) that is a systematic review of MI with MSM. There have been many many more since then as well. You may want to incorporate some of the findings to your introduction.
Overall, the introduction is too short and lacks a lot of detail and information/background to set the foundation and justification for the research analysis. Please provide more detail to give the audience appropriate background knowledge of motivational interviewing and how it has been utilized in MSM populations. -and what’s different about your approach/research study.
Methods:
Motivational Interviewing section: What were the top 10 priority areas? What was the average length of the MI sessions?
Data analysis and coding section: Were the participants purposefully selected or convenience sampled. Both are mentioned in the paper. Please clarify.
Since the study aim is to assess the benefits of MI, the methods section should discuss the exit interview. What were the questions used to assess the benefits of the MI? Please list the (or create a table) of the exit interview questions that were asked.
Results:
Says “most practiced anal sex” however the N=10. That is not “most”. Most would be over half and there were 23 total. Please fix that language.
Discussion:
“However, there was no consistency of condom use over time. As in a study conducted 289 in China among MSM, there was no significant, consistent change in condom use for oral 290 sex or in the number of sexual partners over time.” There is no citation for this study. Please add. Also, there needs to be more detail about this study. Why do you think they found inconsistent results?
Discussion is very underdeveloped. Re-cap the themes. Then tell, in general, a description of each theme. For each theme, the authors should discuss the benefit of the information uncovered. How does knowing that these themes about the utility of MI improve various aspects of counseling MSM regarding sexual behavior, HIV, and STIs?
How could the findings in this study relate to the real-world practices when it comes to interventions, programs, group therapy, and/or individual therapies that seek to reduce outcomes in MSM?
Again, what does this study contribute to the literature that others have not already?
Author Response
RESPONSE TO REVIEWER 2 COMMENTS
Abstract:
Point 1: Says there is limited research on MSM and motivational interviewing. However, there is a lot of previous research on MSM and motivational interviewing. Please eliminate this statement and instead tell us how this study is different (-maybe with respect to this particular population or some other characteristic or outcome; maybe it’s the fact that this study specifically asked participants about the MI process, as opposed to delineating actual behavioral outcomes -I don’t know, but it needs to be clarified). This issue is further discussed in the Introduction feedback below as well.
Response for point 1: The authors do not say there is limited research in general MI and other preventive intervention. The statement if focusing on assessing the “acceptability of MI”
Introduction:
Point 2: Mentions that several studies have shown positive outcomes for the utility of MI in addressing risky sexual behavior, but only two studies are identified in the citations. Please add more studies to this citation.
Response for point 2: More citations were added.
Point 3: Also, it would be beneficial to outline what some of the positive outcomes are that previous studies have found with motivational interviewing by reviewing two or three previous studies. What outcomes were the previous research authors looking to impact; describe the population they studied; what did they do; and what did they find regarding motivational interviewing and its impact on risky sexual behavior?
Response for point 3: This was outlined.
Point 4: Also identify how your study would be different than previous research, as there is already much previous research on MSM and motivational interviewing. There is a 2011 study (Berg, Ross & Tikkanen, 2011) that is a systematic review of MI with MSM. There have been many many more since then as well. You may want to incorporate some of the findings to your introduction.
Response for point 4: Some of the findings were incorporated.
Point 5: Overall, the introduction is too short and lacks a lot of detail and information/background to set the foundation and justification for the research analysis. Please provide more detail to give the audience appropriate background knowledge of motivational interviewing and how it has been utilized in MSM populations. -and what’s different about your approach/research study.
Response for point 5: Introduction was expanded.
Methods:
Point 6: Motivational Interviewing section: What were the top 10 priority areas? What was the average length of the MI sessions?
Response for point 6: Top 10 priority areas added. Average length of MI was included in paragraph 80 and 141 highlighted in yellow.
Point 7: Data analysis and coding section: Were the participants purposefully selected or convenience sampled. Both are mentioned in the paper. Please clarify.
Response for point 7: Participants were purposefully selected.
Point 8: Since the study aim is to assess the benefits of MI, the methods section should discuss the exit interview. What were the questions used to assess the benefits of the MI? Please list the (or create a table) of the exit interview questions that were asked.
Response for point 8: MI exit interview questions were listed in the data collection methods.
Results:
Point 9: Says “most practiced anal sex” however the N=10. That is not “most”. Most would be over half and there were 23 total. Please fix that language.
Response for point 9: Language corrected.
Discussion:
Point 10: “However, there was no consistency of condom use over time. As in a study conducted 289 in China among MSM, there was no significant, consistent change in condom use for oral 290 sex or in the number of sexual partners over time.” There is no citation for this study. Please add. Also, there needs to be more detail about this study. Why do you think they found inconsistent results?
Response for point 10: Citation added.
Point 11: Discussion is very underdeveloped. Re-cap the themes. Then tell, in general, a description of each theme. For each theme, the authors should discuss the benefit of the information uncovered.
How does knowing that these themes about the utility of MI improve various aspects of counseling MSM regarding sexual behavior, HIV, and STIs? – responded in line 405-408
Response for point 11: added line 325-363. Response in the limitation.
Point 12: How could the findings in this study relate to the real-world practices when it comes to interventions, programs, group therapy, and/or individual therapies that seek to reduce outcomes in MSM? And Point 13: Again, what does this study contribute to the literature that others have not already?
Response for point 12 and 13: responded in line 393-400.
Reviewer 3 Report
A qualitative study of the benefits and utility of brief motivational interviewing to reduce sexually transmitted infections among men who have sex with men
This is an article about the utility of brief motivational interviewing to reduce sti among msm. Although interesting enough, the article has several flaws that impede its publication in the journal. Hence, I recommend rejection. Here is the basis of my recommendation:
1. Introduction is very parsimonious and does not reflect the current state of the art on the subject.
2. Objectives are not clear.
3. The study does not adhere to the Consolidated criteria for reporting qualitative research (COREQ), methodological criteria are not clear.
4. Participant’s excerpts are not clearly explained and do not reflect the supposedly advantage of reducing STI.
5. Discussion is poor and insipient. Implications or limitations are not discussed.
Best wishes.
Author Response
RESPONSE TO REVIEWER 3 COMMENTS
Point 1: Introduction is very parsimonious and does not reflect the current state of the art on the subject.
Response for point 1: Introduction was corrected and expanded.
Point 2: Objectives are not clear.
Response for point 2: Objective outline in paragraph 14 and 15
Point 3: The study does not adhere to the Consolidated criteria for reporting qualitative research (COREQ), methodological criteria are not clear.
Response for point 3: COREQ was downloaded and completed. Additional author descriptors are indicated in the authorship contribution section.
Methodology was corrected.
Point 4: Participant’s excerpts are not clearly explained and do not reflect the supposedly advantage of reducing STI.
Response for point 4: The experts are on acceptability and usefulness of the intervention, which is an advantage to reduce transmission of STIs.
Pont 5: Discussion is poor and insipient. Implications or limitations are not discussed.
Response for point 5: Discussion was corrected and extended. Limitations are included in page 10. Paragraph 395.